# The Impact of SARS-CoV-2 Infection on Glucose Homeostasis in Hospitalized Patients with Pulmonary Impairment

**DOI:** 10.3390/diagnostics15050554

**Published:** 2025-02-25

**Authors:** Oana-Andreea Parlițeanu, Mara-Amalia Bălteanu, Dragoș Cosmin Zaharia, Tudor Constantinescu, Alexandra Maria Cristea, Ștefan Dumitrache-Rujinscki, Andra Elena Nica, Cristiana Voineag, Octavian Sabin Alexe, Emilia Tabacu, Alina Croitoru, Irina Strâmbu, Roxana Maria Nemeș, Beatrice Mahler

**Affiliations:** 1Institutul Național de Pneumoftizologie Marius Nasta, 050159 București, Romania; oana_andreea@yahoo.com (O.-A.P.); zahariadragoscosmin@gmail.com (D.C.Z.); tudor_constantinescu@yahoo.com (T.C.); alexandra.maria.sora@gmail.com (A.M.C.); srujinski@yahoo.com (Ș.D.-R.); emiliatabacu@yahoo.com (E.T.); haulicaalina@yahoo.com (A.C.); istrambu@yahoo.com (I.S.); roxanamarianemes@gmail.com (R.M.N.); beatrice.mahler@umfcd.ro (B.M.); 2Department of Pneumology, Universitatea de Medicină și Farmacie Carol Davila, 050474 Bucrești, Romania; dr.andranica@yahoo.com; 3Department of Diabetes, Universitatea Dunărea de Jos, 800201 Galați, Romania; voineag.cristiana@gmail.com (C.V.); octavsabin@yahoo.com (O.S.A.)

**Keywords:** blood glucose, type 2 diabetes mellitus, glycemic imbalance, SARS-CoV-2 infection, pulmonary impairment

## Abstract

**Background and Objectives:** We conducted a retrospective observational study to evaluate the impact of elevated blood glucose levels in patients with SARS-CoV-2 infection and a prior diagnosis of diabetes mellitus (DM) or newly diagnosed hyperglycemia. **Materials and Methods:** This study analyzed 6065 patients admitted to the COVID-19 departments of the “Marius Nasta” National Institute of Pulmonology in Bucharest, Romania, between 26 October 2020 and 5 January 2023. Of these, 813 patients (13.40%) were selected for analysis due to either a pre-existing diagnosis of DM or hyperglycemia at the time of hospital admission. **Results:** The erythrocyte sedimentation rate (ESR) and C-reactive protein (CRP) levels were elevated in patients with blood glucose levels exceeding 300 mg/dL. These elevations correlated with the presence of respiratory failure and increased mortality rates. Additionally, oxygen requirements were significantly higher at elevated blood glucose levels (*p* < 0.001), with a direct relationship between glycemia and oxygen demand. This was accompanied by lower oxygen saturation levels (*p* < 0.001). Maximum blood glucose levels were associated with the severity of respiratory failure (AUC 0.6, 95% CI: 0.56–0.63, *p* < 0.001). We identified cut-off values for blood glucose at admission (217.5 mg/dL) and maximum blood glucose during hospitalization (257.5 mg/dL), both of which were associated with disease severity and identified as risk factors for increased mortality. **Conclusions:** High blood glucose levels, both at admission and during hospitalization, were identified as risk factors for poor prognosis and increased mortality in patients with SARS-CoV-2 infection, regardless of whether the hyperglycemia was due to a prior diagnosis of DM or was newly developed during the hospital stay. These findings underscore the importance of glycemic control in the management of hospitalized COVID-19 patients.

## 1. Introduction

SARS-CoV-2 affects glucose metabolism through insulin resistance induced by a cytokine storm and the use of corticosteroids [1]. Pro-inflammatory cytokines released during the immune response disrupt insulin signaling, while corticosteroids used in COVID-19 treatments increase blood glucose by reducing insulin sensitivity. These effects can worsen existing diabetes or lead to the development of new-onset diabetes, highlighting the importance of understanding the impact of SARS-CoV-2 on glucose metabolism.

This study investigates the impact of hyperglycemia on respiratory function and the prognosis of hospitalized patients with SARS-CoV-2 infection. The identified glucose thresholds correlated with severe respiratory insufficiency and an increased risk of mortality. These findings confirm the results from the existing literature regarding the importance of glycemic control in the context of COVID-19 infection.

The main objective of this research was to evaluate, through an observational, retrospective study, the impact of SARS-CoV-2 infection on glucose metabolism, both in patients with a pre-existing diagnosis of type 2 diabetes mellitus (T2DM) and in patients who developed disorders of glucose metabolism after contracting COVID-19, a phenomenon that has been quite frequently observed [1].

Our hypothesis is that elevated glucose levels, whether present at admission or occurring during hospitalization, may serve as predictive markers for severe respiratory insufficiency and increased mortality in COVID-19 patients. The primary research question addresses the utility of these glucose thresholds in evaluating the severity of pulmonary impairment and clinical prognosis. Secondary objectives included analyzing potential biochemical risk factors such as hyperglycemia, elevated inflammatory markers, D-dimers, oxygen Saturation, influencing disease severity, negative outcomes, and long-term prognosis. To this end, various anthropometric characteristics and biological components with statistical relevance were evaluated.

Studying glycemic homeostasis in the context of SARS-CoV-2 infection is essential, given that the virus can affect pancreatic beta-cell function, leading to acute metabolic disturbances [2]. These dysfunctions can worsen respiratory insufficiency and increase the risk of severe complications, highlighting the importance of early interventions to improve patient care.

## 2. Materials and Methods

We analyzed the database of the Institute of Pneumology “Marius Nasta” in Bucharest, Romania, covering the period from 26 October 2020 to 5 January 2023. The database included patients admitted to COVID-19 wards 5, 6, and 7, which were dedicated exclusively to patients with SARS-CoV-2 infection. Among the 6065 patients admitted during this period, 813 (13.40%) presented with altered blood glucose levels, either due to a pre-existing diagnosis of diabetes mellitus (DM) or as a result of metabolic disturbances developed during hospitalization. Data for all patients were subsequently analyzed using IBM SPSS v.20 to generate statistical results and interpretations.

This study was approved by the hospital’s Ethics Committee, approval number 23943 from 25 October 2023.

Patient Evaluation:

Patients underwent anthropometric assessments, including measurements of height, weight, and calculation of body mass index (BMI), expressed in kg/m^2^.

Biochemical Parameter Measurements:

The biological parameters assessed included blood oxygen saturation (SaO2), complete blood count (leukocytes, hemoglobin, platelets), D-dimers, C-reactive protein (CRP), renal function markers (urea, creatinine, uric acid, and estimated glomerular filtration rate [eGFR]), erythrocyte sedimentation rate (ESR), fibrinogen, liver function markers (aspartate aminotransferase [AST] and alanine aminotransferase [ALT]), ferritin, lactate dehydrogenase (LDH), and blood glucose levels.

Comorbidity Assessment:

Pre-existing medical conditions, including diabetes mellitus, cardiovascular diseases, chronic neoplastic conditions, and respiratory disorders, were also evaluated.

Diabetes Mellitus Diagnosis and Altered Blood Glucose Values:

Diabetes mellitus was diagnosed based on the international criteria established by the American Diabetes Association (ADA). According to these criteria, DM is defined as a fasting blood glucose level exceeding 126 mg/dL on two separate occasions or a random blood glucose level exceeding 200 mg/dL at any time of day [2].

## 3. Results

Of the 813 patients admitted to the COVID-19 pavilion, 449 were men (55.22%) and 364 were women (44.78%). The average age of the entire group was 67.25 years +/−31.24. Among men, the average age was lower at 67.22 years +/−41.11, while among women, the average age was higher at 68.6 years +/−10.07. These 813 patients either had pre-existing diabetes mellitus or newly developed altered blood glucose levels, representing 13.40% of the total COVID-19 patients admitted to the institute during the specified period.

The mortality rate within the group of 813 patients with diabetes mellitus or altered blood glucose levels was 19.06%. Among the 565 deaths in the entire cohort of 6065 SARS-CoV-2 patients, 115 (27.43%) were associated with T2DM or altered glucose levels, suggesting that elevated blood glucose levels and glycemic imbalance were significant risk factors for mortality in this population. With regard to gender, mortality for women was 17.1% and for men was 20.2%.

Of the 813 patients, 524 (64.45%) had a pre-existing diagnosis of diabetes mellitus prior to admission, while 289 (35.55%) had no prior diagnosis of T2DM and were considered to have developed new-onset metabolic disorders during hospitalization.

Corticosteroid treatment was administered to 92.8% of the total 813 patients during their hospitalization; of these, 76.96% were in the survivor group (*p* < 0.001).

An analysis was conducted to evaluate the relationship between the severity of respiratory failure and various clinical and biological parameters. These included blood glucose levels at admission (expressed in milligrams per deciliter [mg/dL]), oxygen requirements (expressed in liters per minute), administered insulin doses (expressed in international units [IUs]), maximum glucose levels (expressed in mg/dL), erythrocyte sedimentation rate (ESR), fibrinogen levels (expressed in mg/dL), ferritin levels (expressed in mg/dL), and the estimated glomerular filtration rate (eGFR, expressed in milliliters per minute per body surface area of 1.73 square meters [mL/min/1.73 m^2^]).

In Table 1, the relationship between the severity of respiratory insufficiency (RI) and variations in certain clinical and biological parameters is highlighted. Statistical analysis indicates that parameters such as blood glucose at admission, oxygen requirement, maximum blood glucose level, ESR, and insulin dose (UI/mL) show significant differences (*p* < 0.05) across the severity levels of RI, suggesting an association between the progression of RI and inflammatory/metabolic responses. Conversely, parameters such as eGFR, insulin dose, fibrinogen, and ferritin do not exhibit statistically significant variations, being primarily influenced by individual variability.

Table 2 highlights the relationship between blood glucose levels (glucose < 126 mg/dL, glucose < 200 mg/dL, glucose 200–300 mg/dL, and glucose > 300 mg/dL) and variations in certain clinical and biological parameters. Statistical analysis shows that oxygen requirement, ESR, D-dimers, CRP, and SaO2 exhibit statistically significant differences (*p* < 0.05) across the groups.

Oxygen requirement progressively increases with higher glucose levels, reflecting an association between hyperglycemia and the need for more intensive respiratory support.ESR and CRP are significantly elevated in patients with glucose levels >300 mg/dL, suggesting heightened inflammatory processes in severe hyperglycemia.D-dimers significantly rise in cases with glucose >300 mg/dL, indicating increased coagulation activation under severe hyperglycemia conditions.SaO_2_ decreases slightly but significantly in patients with glucose >300 mg/dL, potentially signaling impaired oxygenation.

Parameters such as eGFR, fibrinogen, and ferritin do not show significant variations, likely being influenced by factors other than glucose levels.

Table 3 highlights significant differences between survivors and deceased patients based on various clinical and biological parameters. Mortality is associated with the following:Increased oxygen requirement (*p* < 0.001) and reduced oxygen saturation (*p* = 0.001), indicating severe respiratory insufficiency.Reduced kidney function (lower eGFR, *p* < 0.001) and coagulation activation (elevated D-dimers, *p* = 0.001).Severe systemic inflammation, reflected by elevated CRP (*p* = 0.001) and ferritin levels (*p* = 0.001).Hyperglycemia at admission (*p* = 0.004), associated with a poor prognosis.

These parameters can be used for risk assessment and targeted therapeutic interventions in critical cases.

Table 4 highlights significant differences among mild, moderate, and severe forms of COVID-19 regarding several clinical and biological parameters:Significant parameters (*p* < 0.05):

Oxygen requirement, ESR, CRP, oxygen saturation, blood glucose at admission, and maximum blood glucose levels progressively increase with disease severity. This progression reflects respiratory insufficiency, systemic inflammation, and metabolic dysfunction in severe COVID-19 cases.


o
Non-significant parameters:

eGFR, fibrinogen, D-dimers, ferritin, and minimum blood glucose levels do not exhibit statistically relevant variations between groups, although their values tend to be higher in severe forms of the disease.

These indicators emphasize the importance of clinical monitoring for risk stratification and the effective management of COVID-19 patients.

ROC Analysis Between Maximum Blood Glucose and Severe Respiratory Insufficiency

A Receiver Operating Characteristic (ROC) curve analysis could evaluate the predictive value of maximum blood glucose levels for identifying severe respiratory insufficiency, providing a tool for early intervention and outcome improvement.

The ROC analysis of maximum blood glucose, as seen in Figure 1, indicates a statistically significant association between elevated glucose levels and the severity of respiratory insufficiency. The analyzed parameters highlight the importance of this marker in patient assessment:An AUC value of 0.6 (95% CI: 0.56–0.63) suggests a moderate link between maximum blood glucose and the severity of respiratory insufficiency, with the result being statistically significant (*p* < 0.001).Although the discrimination power is moderate, the result underscores the relevance of maximum blood glucose in risk stratification.The established cut-off value of 257.5 mg/dL links maximum blood glucose with adequate sensitivity, indicating that levels above this threshold are more commonly observed in patients with severe respiratory insufficiency.Maximum blood glucose correctly identifies 66% of patients with severe respiratory insufficiency. This sensitivity level suggests that maximum blood glucose is a useful marker for detecting at-risk patients.While the specificity is lower (49%), it helps identify a substantial proportion of patients with normal levels, reducing the risk of under diagnosing severe cases.

ROC Analysis of Admission Blood Glucose and Severe Respiratory Insufficiency

The ROC analysis highlights the importance of admission blood glucose as a marker associated with the severity of respiratory insufficiency (Figure 2), offering valuable insights for risk stratification and patient management.

The ROC parameters emphasize the following:With an AUC of 0.549 (95% CI: 0.51–0.59), admission blood glucose shows a statistically significant association with severe respiratory insufficiency (*p* = 0.01). This result confirms that admission glucose levels are a relevant parameter for identifying patients with more severe forms of respiratory failure.The cut-off value of 217.5 mg/dL represents a critical threshold, delineating cases with increased risk and indicating a direct relationship between elevated glucose levels at admission and severe respiratory insufficiency.Admission blood glucose correctly identifies 40% of patients with severe respiratory insufficiency, making it a useful marker for detecting high-risk cases.Although the specificity is moderate (29%), this parameter helps reduce the risk of underestimating severity in certain cases, making it valuable when used alongside other clinical markers.

Linear Regression Analysis

The linear regression analysis highlights (Table 5) a significant association between maximum blood glucose levels and several relevant clinical parameters, providing important insights in the context of severe respiratory insufficiency:1.Oxygen Requirement (Respiratory Insufficiency)
A significant positive relationship between oxygen requirement and maximum blood glucose (B = 1.640, *p* = 0.037) suggests that hyperglycemia may represent a metabolic response to severe respiratory insufficiency.The increase in maximum glucose levels in this context may reflect the activation of the stress axis (hypercortisolemia) and systemic inflammatory response.This finding strengthens the role of maximum blood glucose as an indirect marker of respiratory insufficiency severity, making it useful for patient monitoring.
2.eGFR (Renal Function)
A significant negative relationship (B = −0.317, *p* = 0.034) indicates that patients with impaired renal function tend to have higher maximum blood glucose levels.This could be attributed to the accumulation of metabolic and hormonal factors contributing to hyperglycemia, such as insulin resistance or the kidney’s inability to counteract excessive gluconeogenesis.Reduced eGFR could therefore serve as an indicator of metabolic risk and complications associated with respiratory insufficiency.
3.ESR (Systemic Inflammation)
A significant positive relationship between ESR and maximum blood glucose (B = 0.382, *p* = 0.018) underscores the role of chronic and acute inflammation in raising maximum glucose levels.Severe inflammatory processes are known to increase pro-inflammatory cytokines’ production (e.g., IL-6, TNF-α), which disrupts insulin action and contributes to hyperglycemia.This suggests that maximum blood glucose could serve as an indirect marker of systemic inflammation severity.
4.Fibrinogen and Ferritin
Although not statistically significant, their relationship with maximum blood glucose reflects a trend potentially explainable by data variability.Fibrinogen and ferritin are well-established markers of inflammation and oxidative stress, and future studies may confirm a clearer link between these variables and maximum blood glucose.



Clinical Implications
Maximum blood glucose can serve as an integrated indicator of the body’s response to severe stress caused by respiratory insufficiency, renal failure, and systemic inflammation.In practice, monitoring maximum blood glucose alongside associated variables such as oxygen requirement, eGFR, and ESR can help identify patients at higher risk for complications and guide therapeutic decisions.Interventions aimed at glucose control, reducing inflammation, and optimizing respiratory and renal function could improve clinical outcomes.
Future Directions
Additional studies are needed to explore causal relationships between these parameters and validate maximum blood glucose as a prognostic marker in severe respiratory insufficiency.Integrating maximum blood glucose into multidimensional predictive models could provide a valuable tool for managing critically ill patients.



## 4. Discussion

Pancreatic involvement in SARS-CoV-2 infection has been extensively studied, with evidence demonstrating a positive relationship between COVID-19 and the de novo onset of diabetes mellitus or glucose metabolism impairment, even in patients with no prior history of diabetes [3]. Antonio Ceriello and colleagues described mechanisms by which COVID-19 affects pancreatic function and reiterated that hyperglycemia, even in patients without a history of diabetes, is a significant risk factor that complicates the progression of SARS-CoV-2 infection [3,4,5,6,7].

In severe cases of SARS-CoV-2 infection, secondary pancreatic dysfunction has been observed due to the cellular expression of angiotensin-converting enzyme 2 (ACE-2). This leads to beta-cell destruction and a consequent reduction in insulin secretion, resulting in elevated blood glucose levels, exacerbating pre-existing diabetes, or triggering de novo diabetes [5].

Hyperglycemia in SARS-CoV-2 patients has been shown to be a risk factor for disease severity and increased mortality [6,7]. Studies have also reported that hyperglycemia may arise from an acute exacerbation of chronic disease, a newly developed acute episode in a previously diagnosed diabetic patient, or as a part of the body’s response to acute pathology in non-diabetic patients [8,9,10]. Some patients develop a specific form of hyperglycemia referred to as “stress-induced diabetes” [10].

In our study, over one-third of patients (35.55%) presented with elevated blood glucose levels and newly diagnosed diabetes, despite having no prior diagnosis of the condition. These findings align with the existing literature, supporting the hypothesis that SARS-CoV-2 impacts pancreatic beta-cell function.

A blood glucose threshold of 217.5 mg/dL in our study was identified as a critical value separating high-risk cases, correlating directly with elevated glucose levels at admission. Higher glucose levels were associated with greater oxygen requirements, indicating a link between hyperglycemia and severe respiratory insufficiency, which exacerbates respiratory dysfunction through metabolic disturbances.

We demonstrated, with an AUC value of 0.549 (95% CI: 0.51–0.59), that elevated glucose levels at admission were associated with respiratory insufficiency severity (*p* = 0.01). This result corroborates the findings in the literature, emphasizing that hyperglycemia in SARS-CoV-2 patients is a significant risk factor for severe disease.

Regarding mortality, our study data confirm that elevated blood glucose is a risk factor for higher mortality rates. In our cohort, mortality was 27.43% (N = 115), and elevated admission glucose levels were positively correlated with unfavorable outcomes (*p* = 0.004).

Sardu and colleagues described persistent elevations in D-dimer levels throughout hospitalization in SARS-CoV-2 patients with elevated blood glucose [11,12,13]. Similarly, in our study, D-dimer levels were elevated in patients with glucose levels exceeding 300 mg/dL, consistent with prior findings.

Sardu et al. also highlighted increased inflammatory processes in SARS-CoV-2 patients with hyperglycemia [14,15,16,17]. Their study demonstrated that patients with high glucose levels were more likely to exhibit elevated pro-inflammatory cytokines, particularly interleukin-6 (IL-6). While our study did not assess pro-inflammatory cytokines, systemic inflammation markers such as ferritin, ESR, and CRP were significantly elevated in patients with glucose levels >300 mg/dL, reflecting an enhanced inflammatory response due to hyperglycemia.

Hyperglycemia has long been studied as a predictor of poor outcomes during hospitalization in patients with acute conditions or exacerbated chronic diseases [18]. Consistent with this, our findings confirm that hyperglycemia in SARS-CoV-2 infection is a negative prognostic factor.

In COVID-19 patients, hyperglycemia at admission resulted in hypercoagulability, reflected by elevated D-dimer levels, and a heightened systemic inflammatory response, as evidenced by increased ESR, CRP, and ferritin levels [19,20,21,22,23,24,25]. Patients with glucose levels >300 mg/dL in our study exhibited statistically significant increases in D-dimer, ESR, and ferritin, aligning with previous research.

To prevent poor outcomes in patients with hyperglycemia at admission or during hospitalization, the American Diabetes Association (ADA) recommends maintaining glucose levels within 7.77–9.99 mmol/L (138.6–179.82 mg/dL) for patients with acute infections and below 9.99 mmol/L (179.82 mg/dL) for septic patients [26,27]. In our study, the threshold values for glucose levels at admission (217.5 mg/dL) and for predicting respiratory insufficiency (257.5 mg/dL) were higher than the ADA recommendations, further emphasizing the prognostic value of hyperglycemia in acute SARS-CoV-2 infection [14].

From the literature, we also observed that our findings regarding the pro-thrombotic state (elevated D-dimers) and oxidative stress response (elevated systemic inflammatory markers) in hyperglycemia are consistent with previously reported risk factors for SARS-CoV-2 patients with acute infection [18,19,20,21]. The interplay between hyperglycemia, inflammation, and microvascular dysfunction creates a vicious cycle that increases oxygen demand and impairs oxygen delivery and utilization. The effective management of hyperglycemia through glycemic control, anti-inflammatory therapies, and supportive care targeting oxygenation is crucial to improve outcomes in COVID-19 patients [11].

The lack of significant correlations between fibrinogen, ferritin, and maximum blood glucose levels is likely attributable to a combination of biological variability, including pre-existing conditions such as cardiovascular diseases, which are highly prevalent in diabetic patients, and corticosteroid treatments, which can suppress hyperinflammatory states and reduce variability in fibrinogen and ferritin levels [3].

This study has several limitations. First, its retrospective design inherently limits the ability to establish causal relationships between hyperglycemia and adverse outcomes. In COVID-19 patients, hyperglycemia could be seen as a marker of severity rather than a direct contributor to the poor outcomes. Second, potential confounding factors, such as pre-existing comorbidities (e.g., cardiovascular diseases, obesity) and treatment protocols (e.g., corticosteroid use), may have influenced the observed associations. Third, the lack of long-term follow-up data precludes the assessment of the chronic effects of hyperglycemia or its impact on recovery and post-acute COVID-19 syndrome. Also, one of the limitations of this study was the lack of glucose-lowering therapies used for these patients during hospitalization. Finally, this study was conducted in a single center from Romania, which may limit the generalization of the findings to broader populations.

The patients in our study also presented with other comorbidities, such as arterial hypertension, various neoplasms, heart failure, bronchial asthma, chronic kidney disease, depression, or dementia. However, these comorbidities did not have a statistically significant impact on mortality. Therefore, we decided not to include them in this article, choosing to focus on the variables that showed statistical significance; this was another limitation of our research.

## 5. Conclusions


Maximum Blood Glucose and Respiratory Insufficiency
oMaximum blood glucose is associated with the severity of respiratory insufficiency, serving as a simple and accessible marker that can support clinical decision-making.oThe identified threshold of 257.5 mg/dL provides guidance for risk stratification and prioritizing patients based on the severity of their condition.oThe statistical significance of the results (*p* < 0.001) reinforces the role of maximum blood glucose as a relevant parameter in the evaluation of respiratory insufficiency.



These findings suggest that maximum hyperglycemia could be integrated into a multidimensional evaluation model, offering valuable insights for patient management.


2.Admission Blood Glucose
oAdmission blood glucose is an accessible and easy-to-measure indicator that can guide clinical decisions and stratify patients by the severity of respiratory insufficiency.oIts statistical significance (*p* = 0.01) emphasizes its importance in identifying severe cases of disease, and the threshold of 217.5 mg/dL provides a practical basis for initial risk assessment.oWhile not an exclusive marker, admission blood glucose makes a valuable contribution within a multidimensional evaluation framework, offering early insights into the severity of respiratory insufficiency and supporting effective clinical management.
3.Comparative Performance of the Two Markers
oMaximum blood glucose demonstrates a slightly higher AUC compared to admission blood glucose, suggesting that it is a better marker for stratifying respiratory insufficiency severity.oMaximum blood glucose achieves a better balance between sensitivity and specificity, making it more useful for detecting severe respiratory insufficiency.oMaximum blood glucose outperforms admission blood glucose in identifying patients with severe respiratory insufficiency.
4.Statistical Significance of Both Markers
oBoth markers show statistical significance (*p* < 0.05), confirming their relevance in assessing the severity of respiratory insufficiency. However, maximum blood glucose demonstrates stronger significance (*p* < 0.001) compared to admission blood glucose (*p* = 0.01).




Overall Conclusion


Maximum blood glucose appears to be more useful for evaluating the severity of respiratory insufficiency, while admission blood glucose complements this assessment during the early stages of patient management. Together, these markers provide a robust foundation for risk stratification and clinical decision-making.

## Figures and Tables

**Figure 1 diagnostics-15-00554-f001:**
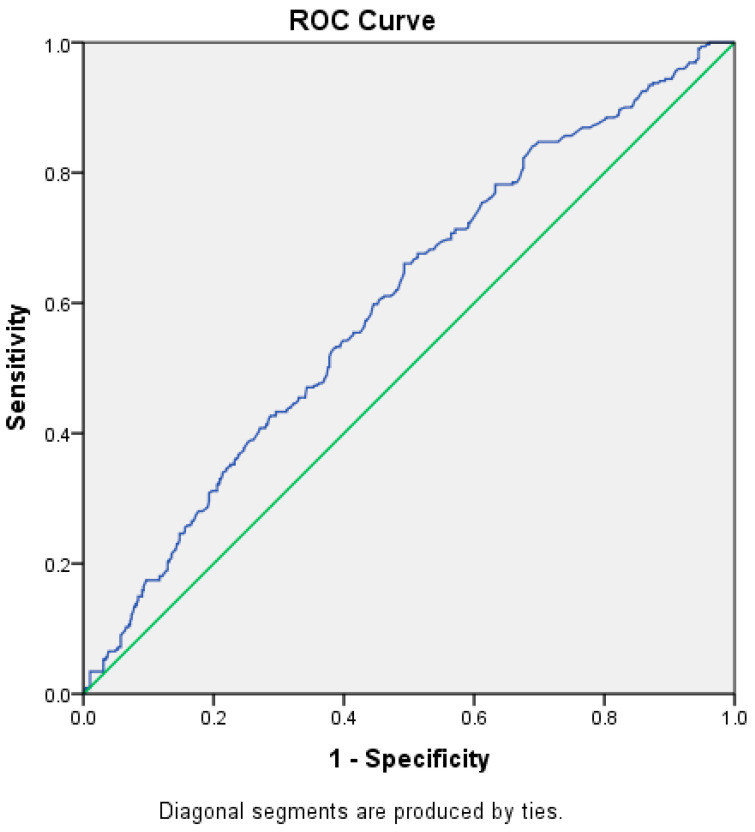
ROC Analysis of Maximum Blood Glucose and Severe Respiratory Insufficiency.

**Figure 2 diagnostics-15-00554-f002:**
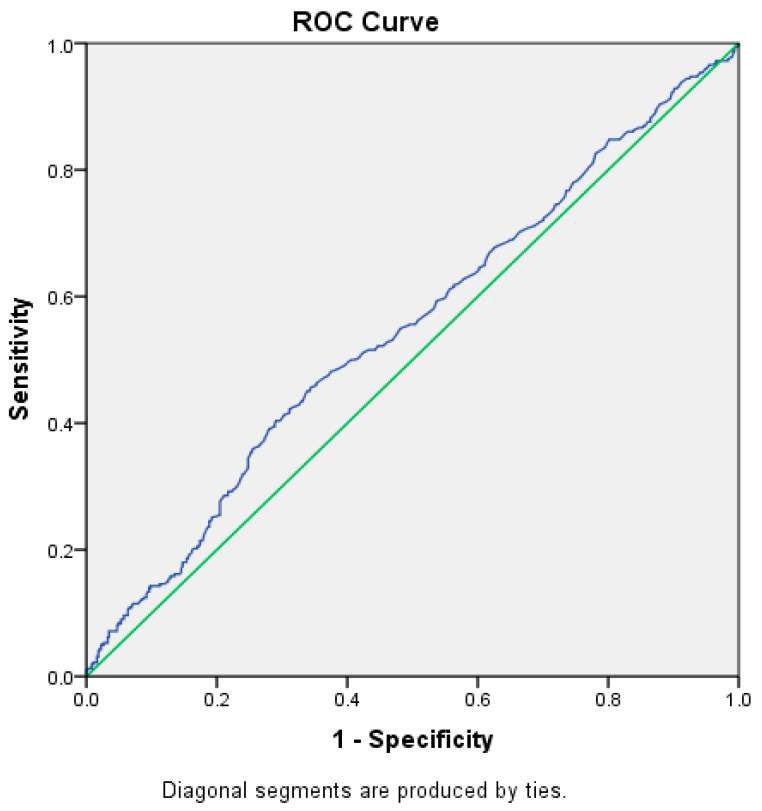
ROC Analysis of Admission Blood Glucose and Severe Respiratory Insufficiency.

**Table 1 diagnostics-15-00554-t001:** The relationship between the severity of respiratory insufficiency and clinical and biological variations.

	Mild Respiratory Failure		Moderate Respiratory Failure		Severe Respiratory Failure		Total		
	Mean	Std. Deviation	Mean	Std. Deviation	Mean	Std. Deviation	Mean	Std. Deviation	*p*-Value
Admission glucose value (mg/dL)	202.04	109.30	198.73	90.17	219.73	117.86	207.16	105.94	0.039
O_2_ requirement	2.02	3.45	5.53	4.46	11.02	8.10	6.72	6.97	0.001
eGFR (mL/min/1.73 m^2^)	72.44	32.75	77.41	35.43	72.91	32.04	74.55	33.31	0.296
Dose UI/ML	5593.51	2138.93	5853.76	2641.38	6817.90	4206.35	6178.16	3320.25	0.001
Dose	38.36	24.39	40.82	46.70	37.05	17.28	38.91	32.23	0.957
Maximum blood glucose value (mg/dL)	281.06	184.03	296.51	119.99	325.21	131.07	301.62	141.48	0.001
ESR	56.71	32.02	62.44	34.89	74.96	33.79	65.97	34.54	0.001
Fibrinogen (mg.dL)	457.20	119.74	474.72	114.54	462.58	166.83	466.33	138.71	0.509
Feritin (mg/dL)	890.32	1134.37	1524.20	5163.53	976.64	2445.97	1151.05	3545.82	0.177

**Table 2 diagnostics-15-00554-t002:** The relationship between glycemic levels and clinical and biological variations.

	Glucose < 126 mg/dL		Glucose < 200 mg/dL		Glucose 200–300 mg/dL		3 = Glucose > 300 mg/dL		
	Mean	Std. Deviation	Mean	Std. Deviation	Mean	Std. Deviation	Mean	Std. Deviation	*p*-Value
O_2_ requirement	5.28	6.96	6.34	6.69	7.56	6.83	8.06	7.55	0.00
eGFR (mL/min/1.73 m^2^)	71.66	32.35	75.65	32.45	77.53	35.70	70.46	32.38	0.17
ESR	65.27	35.45	63.29	34.79	65.65	31.85	74.23	36.06	0.03
Fibrinogen (mg/dL)	456.10	121.37	471.25	152.83	471.73	129.09	457.71	135.93	0.57
D-DIMERI (mg/dL)	2056.61	4808.40	2050.55	5998.22	1763.58	3939.55	5011.33	21,799.04	0.02
CRP	85.66	78.12	101.68	95.93	102.60	99.21	127.65	112.51	0.01
Feritin (mg/dL)	991.69	3335.13	938.99	1622.36	1222.12	3606.13	1812.19	6394.62	0.16
SaO_2_	90.63	10.51	91.23	7.26	89.38	10.06	88.85	10.41	0.00

**Table 3 diagnostics-15-00554-t003:** Differences between survivors and deceased patients in relation to clinical and biological parameters.

	Survivor		Dead		Total		
	Mean	Std. Deviation	Mean	Std. Deviation	Mean	Std. Deviation	*p*-Value
O_2_ requirement	5.65	5.64	11.48	9.79	6.72	6.97	<0.001
eGFR (mL/min/1.73 m^2^)	80.43	29.87	49.44	35.71	74.55	33.31	<0.001
ESR	65.63	34.83	67.91	33.10	65.97	34.54	0.323
Fibrinogen (mg/dL)	460.69	135.30	491.89	150.35	466.33	138.71	0.023
D-DIMERI (mg/dL)	1676.67	5066.30	5687.35	19764.65	2438.77	9861.04	0.001
CRP (mg/dL)	88.21	87.23	167.04	111.04	102.93	97.11	0.001
Feritin (mg/dL)	756.25	658.38	2822.16	7794.72	1151.05	3545.82	0.001
SaO_2_	91.23	8.24	85.89	11.90	90.27	9.23	0.001
Admission glucose value (mg/dL)	201.87	97.65	230.89	133.78	207.16	105.94	0.004

**Table 4 diagnostics-15-00554-t004:** Differences between COVID-19 severity levels in relation to clinical and biological parameters.

	Mild COVID-19		Medium COVID-19		Severe COVID-19		Total		
	Mean	Std. Deviation	Mean	Std. Deviation	Mean	Std. Deviation	Mean	Std. Deviation	*p*-Value
O_2_ requirement	2.64	3.91	3.64	4.97	8.34	7.32	6.72	6.97	<0.001
eGFR (mL/min/1.73 m^2^)	75.51	29.44	78.85	30.32	72.97	34.54	74.55	33.31	0.118
ESR	55.66	31.80	57.04	34.29	70.21	34.15	65.97	34.54	<0.001
Fibrinogen (mg/dL)	444.87	121.77	454.17	174.87	472.45	124.01	466.33	138.71	0.239
D-DIMERI (mg/dL)	747.23	904.60	1904.76	6494.72	2799.43	11248.44	2438.77	9861.04	0.414
CRP (mg/dL)	62.02	74.23	84.99	86.01	112.88	101.13	102.93	97.11	<0.001
Feritin (mg/dL)	701.03	725.12	781.87	980.10	1321.94	4202.62	1151.05	3545.82	0.282
SaO_2_	94.81	3.63	93.43	5.34	88.62	10.27	90.27	9.23	<0.001
Admission glucose value (mg/dL)	187.92	117.50	194.82	94.17	212.90	108.54	207.16	105.94	0.037
Maximum blood glucose level (mg/dL)	239.88	123.57	285.88	118.66	313.07	149.23	301.62	141.48	0.001
Minimum blood glucose level (mg/dL)	129.10	48.58	150.55	72.46	147.66	92.86	147.35	85.99	0.473

**Table 5 diagnostics-15-00554-t005:** Linear regression.

Coefficients ^a^								
Model		Unstandardized Coefficients		Standardized Coefficients	t	Sig.	95.0% Confidence Interval for B	
		B	Std. Error	Beta			Lower Bound	Upper Bound
	O_2_ requirement	1.640	0.783	0.083	2.095	0.037	0.103	3.178
	eGFR(mL/min/1.73 m^2^)	−0.317	0.149	−0.075	−2.124	0.034	−0.610	0.024
	ESR	0.382	0.162	0.096	2.363	0.018	0.065	0.699
	Fibrinogen (mg/dL)	−0.063	0.040	−0.064	−1.564	0.118	−0.143	0.016
	Feritin (mg/dL)	0.002	0.002	0.029	0.734	0.463	−0.003	0.007

^a^ Dependent variable: GLI MAX.

## Data Availability

The original contributions presented in this study are included in the article. Further inquiries can be directed to the corresponding author.

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
