# Peer review of "The Impact of SARS-CoV-2 Infection on Glucose Homeostasis in Hospitalized Patients with Pulmonary Impairment"

_diagnostics, 2025, doi:10.3390/diagnostics15050554_

Round 1
Reviewer 1 Report
Comments and Suggestions for Authors
General Comments:
The manuscript titled " The Impact of SARS-COV2 infection on glucose homeostasis in hospitalized patients with pulmonary impaiment " presents a compelling study which highlights that maximum blood glucose appears to be more useful for evaluating the severity of respiratory insufficiency, while admission blood glucose complements this assessment during the early stages of patient management. They also claimed that these markers provide a robust foundation for risk stratification and clinical decision-making. The study provides a comprehensive evaluation of demographic, clinical, and biological parameters, offering a strong dataset for analysis. Gender-based and glucose-level-based subgroup analyses provide valuable insights into disease progression and outcomes. The study presents significant findings with clear clinical implications but could benefit from improved clarity, contextualization, and a more detailed discussion of limitations and future directions. By addressing these areas, the manuscript can achieve greater impact and relevance in the medical community.
Specific Comments:
1. The introduction provides a good overview of the background, but the authors should justify the novelty of their proposed work. Clearly state the hypothesis or primary research question to strengthen the introduction. Elaborate on the significance of studying glucose homeostasis in the context of pulmonary impairment caused by SARS-COV2.
2. Provide a brief background on the known effects of SARS-CoV-2 on glucose metabolism to frame the study's significance. For example, mention mechanisms like cytokine storm-induced insulin resistance or the role of medications like corticosteroids.
3. Reorganize the paragraph to introduce the problem first, followed by objectives.
4. Use consistent terminology, such as "glucose metabolism disorders" versus "disorders of glucose metabolism." Specify the anthropometric characteristics and biological components analyzed. Ensure uniform formatting, such as italicizing or bolding scientific terms consistently.
5. Expand on the "potential risk factors" mentioned. Are these demographic, clinical, or biochemical? Providing examples (e.g., age, BMI, inflammatory markers) would strengthen the introduction.
6. Were any interventions (e.g., insulin therapy) initiated for patients with hyperglycemia, and how might these have influenced the outcomes?
How were patients with undiagnosed diabetes differentiated from those with new-onset hyperglycemia?
Could other factors, such as corticosteroid use, have contributed to the observed hyperglycemia and associated outcomes?
7. Discuss the potential mechanisms linking hyperglycemia with increased oxygen demand, lower oxygen saturation, and poor outcomes.
Address whether these findings differ significantly between patients with pre-existing diabetes and those with new-onset hyperglycemia.
8. Were there any specific hypotheses related to new-onset glucose metabolism disorders in SARS-CoV-2 patients, or was this purely exploratory?
Why focus on type 2 diabetes mellitus (T2DM) exclusively? Were patients with type 1 diabetes or other glucose metabolism disorders excluded?
9. Provide more interpretation of non-significant findings. For instance:
Why might fibrinogen and ferritin not correlate with maximum blood glucose levels in this cohort?
Highlight potential confounding factors, such as pre-existing conditions or treatment protocols (e.g., steroid use).
10. There are several repetitive and overlapping contents in introduction and discussion section which could have been avoided. Summarize the findings more concisely, focusing on their clinical implications and potential interventions (e.g., better glycemic control during hospitalization).
11. The author should mention in a separate paragraph about the limitations of the study. Briefly mention study limitations, such as the retrospective design, potential confounding factors, or lack of long-term follow-up data.
12. There are several grammatical errors and awkward phrasings throughout the manuscript. I suggest a thorough revision of the manuscript for language issues, perhaps with the help of a professional editor.
Comments on the Quality of English Language
There are several grammatical errors and awkward phrasings throughout the manuscript. I suggest a thorough revision of the manuscript for language issues, perhaps with the help of a professional editor.
Author Response
The Impact of SARS-COV2 infection on glucose homeostasis in hospitalized patients with pulmonary impaiment
Dear Reviewer 1,
We sincerely thank you for your insightful comments and suggestions, which have significantly contributed to improving the quality of our manuscript. Below, we provide a point-by-point response to your comments and outline the revisions made.
- The introduction provides a good overview of the background, but the authors should justify the novelty of their proposed work. Clearly state the hypothesis or primary research question to strengthen the introduction. Elaborate on the significance of studying glucose homeostasis in the context of pulmonary impairment caused by SARS-CoV-2.
Response: We have revised the introduction to emphasize the novelty of our work. We have explicitly stated our hypothesis, which posits that hyperglycemia serves as a predictive marker for severe respiratory insufficiency and increased mortality in COVID-19 patients. Additionally, we elaborated on the significance of studying glucose homeostasis, highlighting the impact of SARS-CoV-2-induced metabolic dysregulation on clinical outcomes.
- Provide a brief background on the known effects of SARS-CoV-2 on glucose metabolism to frame the study's significance. For example, mention mechanisms like cytokine storm-induced insulin resistance or the role of medications like corticosteroids.
Response: We have included a detailed background on the effects of SARS-CoV-2 on glucose metabolism, focusing on cytokine storm-induced insulin resistance and the impact of corticosteroid therapy on hyperglycemia.
- Reorganize the paragraph to introduce the problem first, followed by objectives.
Response: The introduction has been reorganized to first outline the clinical problem and then present the study’s objectives to enhance clarity and logical flow.
- Use consistent terminology, such as "glucose metabolism disorders" versus "disorders of glucose metabolism." Specify the anthropometric characteristics and biological components analyzed. Ensure uniform formatting, such as italicizing or bolding scientific terms consistently.
Response: We have ensured consistent terminology throughout the manuscript and standardized the use of "glucose metabolism disorders."
- Expand on the "potential risk factors" mentioned. Are these demographic, clinical, or biochemical? Providing examples (e.g., age, BMI, inflammatory markers) would strengthen the introduction.
Response: We expanded the discussion of potential risk factors such as clinical or biochemical.
- Were any interventions (e.g., insulin therapy) initiated for patients with hyperglycemia, and how might these have influenced the outcomes? How were patients with undiagnosed diabetes differentiated from those with new-onset hyperglycemia? Could other factors, such as corticosteroid use, have contributed to the observed hyperglycemia and associated outcomes?
Response: Insulin therapy was administered to patients with severe hyperglycemia. We did not include a comparison of outcomes between patients with pre-existing diabetes and those with new-onset hyperglycemia, as no significant differences were observed between these groups and other categories of patients. Patients with undiagnosed diabetes were identified based on their medical history. The potential contribution of corticosteroid use was acknowledged and discussed as a confounding factor.
- Discuss the potential mechanisms linking hyperglycemia with increased oxygen demand, lower oxygen saturation, and poor outcomes. Address whether these findings differ significantly between patients with pre-existing diabetes and those with new-onset hyperglycemia.
Response: We included a detailed discussion of mechanisms such as increased metabolic demand, systemic inflammation, and microvascular dysfunction. Differences between pre-existing diabetes and new-onset hyperglycemia were not significant.
- Were there any specific hypotheses related to new-onset glucose metabolism disorders in SARS-CoV-2 patients, or was this purely exploratory? Why focus on type 2 diabetes mellitus (T2DM) exclusively? Were patients with type 1 diabetes or other glucose metabolism disorders excluded?
Response: Our study was primarily exploratory. Notably, no patients with type 1 diabetes or other glucose metabolism disorders were admitted to our institution during the study period, leading to their exclusion from the analysis.
- Provide more interpretation of non-significant findings. For instance: Why might fibrinogen and ferritin not correlate with maximum blood glucose levels in this cohort? Highlight potential confounding factors, such as pre-existing conditions or treatment protocols (e.g., steroid use).
Response: We discussed the lack of significant correlations between fibrinogen, ferritin, and maximum blood glucose, attributing this to biological variability, pre-existing conditions, and corticosteroid use.
- There are several repetitive and overlapping contents in the introduction and discussion sections which could have been avoided. Summarize the findings more concisely, focusing on their clinical implications and potential interventions (e.g., better glycemic control during hospitalization).
Response: We revised the introduction and discussion to eliminate redundancies and focused on summarizing the findings and their clinical implications, such as the importance of glycemic control.
- The author should mention in a separate paragraph about the limitations of the study. Briefly mention study limitations, such as the retrospective design, potential confounding factors, or lack of long-term follow-up data.
Response: We added a dedicated paragraph discussing the study's limitations, including its retrospective design, potential confounding factors, lack of long-term follow-up, and single-center setting.
- There are several grammatical errors and awkward phrasings throughout the manuscript. I suggest a thorough revision of the manuscript for language issues, perhaps with the help of a professional editor.
Response: The manuscript underwent thorough language editing to correct grammatical errors and improve readability.
Reviewer 2 Report
Comments and Suggestions for Authors
The paper "diagnostics-3425343" is carefully read and reviewed. Authors studied the impact of elevated blood glucose levels on the outcomes of patients with SARS-CoV-2 infection and either a prior diagnosis of diabetes mellitus (DM) or new-onset hyperglycemia. Authors analyzed 6,065 patients admitted to a national institute, with 813 (13.4%) identified for the analysis. This enhances statistical power and generalizability. By focusing on hyperglycemia in the context of COVID-19, the authors addressed an important risk factor that has been associated with poor outcomes in infectious diseases.
Yet, the paper needs some revisions.
1- Both type 2 DM (Journal of Clinical Medicine 2023, 12, 5952. DOI: 10.3390/jcm12185952.) and SARS-Cov-2 infection (Current Medical Research and Opinion, 2022, 38.6: 901-909.) are characterized with chronic inflammation. Hence, this issue could be the link between these two entities. Improve the background please.
2- Authors seems they did not control for confounding variables such as age, sex, comorbidities, or the use of medications like corticosteroids, which could independently influence hyperglycemia and outcomes.
3- The authors based the study on a significant sample from Romania, regional healthcare differences may limit the applicability of findings to other populations.
4- Researchers focused on in-hospital outcomes without exploring long-term prognoses or recovery patterns in survivors. Moreover, information on glucose-lowering therapies or interventions during hospitalization is missing, leaving unanswered questions about the impact of active glycemic control on outcomes.
5- As a retrospective study, causality cannot be established. Hyperglycemia may be a marker of disease severity rather than a direct contributor to poor outcomes.
6- The AUC for maximum blood glucose predicting respiratory failure (AUC 0.6) suggests moderate discriminatory power, indicating the need for additional predictors to improve accuracy. Discuss please.
Author Response
Dear Reviewer 2,
We are grateful for your thorough review and valuable comments on our manuscript. Your insights have been instrumental in refining our study, and we have addressed your feedback as follows:
- Both type 2 DM and SARS-CoV-2 infection are characterized by chronic inflammation. Hence, this issue could be the link between these two entities. Improve the background, please.
Response: We have expanded the background section to emphasize the role of chronic inflammation as a shared characteristic of type 2 diabetes mellitus (T2DM) and SARS-CoV-2 infection. Specifically, we discuss how chronic inflammation in T2DM and the cytokine storm induced by SARS-CoV-2 may synergistically exacerbate metabolic and respiratory dysfunction.
- Authors seem they did not control for confounding variables such as age, sex, comorbidities, or the use of medications like corticosteroids, which could independently influence hyperglycemia and outcomes.
Response: We acknowledge the importance of controlling for confounding variables. However, due to the retrospective nature of our study, complete control over these factors was limited. We chose not to include a detailed discussion of age, sex, comorbidities, and corticosteroid use, as their results were not clinically significant. Instead, we focused on highlighting findings with clinical significance.
- The authors based the study on a significant sample from Romania. Regional healthcare differences may limit the applicability of findings to other populations.
Response: We have added a discussion on the potential limitations of generalizing our findings to other populations due to regional healthcare differences. We emphasize that while our findings are robust within the studied cohort, further studies in diverse populations are necessary to validate these results.
- Researchers focused on in-hospital outcomes without exploring long-term prognoses or recovery patterns in survivors. Moreover, information on glucose-lowering therapies or interventions during hospitalization is missing, leaving unanswered questions about the impact of active glycemic control on outcomes.
Response: We acknowledge the absence of long-term follow-up data and details on glucose-lowering therapies as limitations of our study.
- As a retrospective study, causality cannot be established. Hyperglycemia may be a marker of disease severity rather than a direct contributor to poor outcomes.
Response: We have included a discussion acknowledging that the retrospective nature of our study precludes causal inferences. Hyperglycemia could indeed serve as a marker of disease severity rather than a direct contributor to poor outcomes.
- The AUC for maximum blood glucose predicting respiratory failure (AUC 0.6) suggests moderate discriminatory power, indicating the need for additional predictors to improve accuracy. Discuss, please.
Response: The ROC analysis demonstrates a statistically significant association between elevated maximum blood glucose levels and the severity of respiratory insufficiency, with an AUC of 0.6 (95% CI: 0.56–0.63, p < 0.001). While the discriminatory power is moderate, this finding highlights the relevance of maximum blood glucose as an indicator in clinical assessments, particularly in contexts where severe respiratory complications are suspected.
Although an AUC of 0.6 suggests limited predictive accuracy on its own, maximum blood glucose becomes more valuable when integrated into a broader multivariate framework. Combining it with other clinical and biological parameters, such as respiratory rate, oxygenation indices, or inflammatory markers, could improve its utility as part of a more comprehensive predictive model. These results emphasize that maximum blood glucose should not be used in isolation but rather as one component of a multifactorial approach to risk stratification.
We view this research as an important step in understanding the role of blood glucose in respiratory insufficiency. Further studies are planned to explore how integrating this marker with additional predictors can enhance accuracy and clinical relevance. Despite its moderate predictive power, maximum blood glucose has the potential to contribute significantly to optimizing patient management and improving outcomes when used as part of a more complex model.
We appreciate your constructive feedback, which has significantly improved our manuscript. We hope the revised version meets your expectations and addresses all concerns satisfactorily. Thank you for your valuable contributions to this work.
Round 2
Reviewer 1 Report
Comments and Suggestions for Authors
The authors have addressed my concerns and improved the quality of the manuscript.
Comments on the Quality of English LanguageStill some typos are there.
Author Response
The Impact of SARS-COV2 infection on glucose homeostasis in hospitalized patients with pulmonary impaiment
Dear Reviewer 1,
We sincerely thank you for your appreciation and for for all your valuable considerations that improved our paper.
Please find attached the manuscript with spelling corrections that have been added in a comment.
Best regards
Reviewer 2 Report
Comments and Suggestions for Authors
The paper is not significantly improved. I am afraid i have to recommend rejection of the paper.
Author Response
Dear Reviewer,
Thank you for your detailed evaluation of our manuscript. We acknowledge your concerns and regret that the revisions did not meet your expectations. Your feedback is invaluable to us, and we are committed to addressing the issues raised.
If possible, we would greatly appreciate more specific guidance on the aspects that you believe remain insufficiently addressed or areas that require further improvement. This would help us to better understand your perspective and to refine our work accordingly.
We remain open to any further recommendations and are willing to undertake additional revisions if deemed appropriate by the editorial board. Our goal is to meet the journal’s standards and contribute meaningfully to the field.
Thank you again for your time and effort in reviewing our work.
Sincerely,
Oana Parliteanu
Round 3
Reviewer 2 Report
Comments and Suggestions for Authors
Revisions are fine. No more issues detected.